# Obtention and Characterization of GO/Epoxy and GO-GPTMS/Epoxy Nanocompounds with Different Oxidation Degrees and Ultrasound Methods

Areli Marlen Salgado-Delgado, Elizabeth Grissel González-Mondragón, Ricardo Hernández-Pérez, René Salgado-Delgado, José Alfonso Santana-Camilo and Alfredo Olarte-Paredes *

Tecnológico Nacional de México, Instituto Tecnológico/IT de Zacatepec, Calzada Tecnológico 27, Centro, Zacatepec de Hidalgo 62780, Morelos, Mexico
* Correspondence: alfredo.op@zacatepec.tecnm.mx

**Abstract:** This work reports the obtention of nanocompounds from epoxy resin (EP) with graphenes at three different oxidation degrees (GO1, GO2, and GO3), functionalized with 3-glycidyloxypropyl trimethoxysilane (GPTMS), and three different graphene concentrations (1%, 2%, and 3%). The aim is to improve GO compatibility in EP and obtain a nanocompound with synergistic properties. Ultrasonic bath was used to disperse the GO, a factor in the effective interaction between GO and the polymeric matrix. The nanocompounds were characterized by FTIR, SEM, and mechanical tension testing. The FTIR analysis evidenced stretching bonds created during the functionalization of graphene oxide (GO) with the silane (GPTMS); they are characteristic Si-O-Si and Si-O-C at 1000 and 1085 cm$^{-1}$, respectively. There was a difference between GO and GO-GPTMS nanocompounds regarding the formation of these signals. The SEM micrographs showed morphological changes when GO was added: the smooth fracture surface of EP became rougher. During tension testing, Young's modulus (2.09 GPa) of GO2-GPTMS/epoxy nanocompounds (1% weight GO) increased by 35% while their resistance to traction (98.71 MPa) grew by 52%; both were higher than in pure EP. In conclusion, the variables studied (oxidation degrees and silanization) significantly affect the mechanical properties studied.

**Keywords:** epoxy resin; graphene oxide; silane coupling agent; nanocompound; ultrasonic bath





## 1. Introduction

Nanocompounds are compound materials where the matrix material is reinforced by one or more separated nanomaterials to improve their properties [1,2]. Polymers as epoxy resin (EP), nylon, and polyepoxide polyetherimide; ceramics such as alumina, glass, and porcelain; and metals as iron, titanium, and magnesium are the most common matrices in nanocompounds [3,4]. Polymeric nanocompounds combine the excellent properties of the polymeric matrix and those of nanofillers in the material design. EPs have been widely used as matrices for coatings and nanocompounds due to their excellent mechanical properties and chemical resistance [5–7]. However, EPs have free volumes in their structures, making them permeable to corrosive agents such as water, oxygen, and destructive ions as Cl$^-$ and H$^+$. The latter can penetrate the interface between the coating and the substrate through micropores that are inevitably produced due to solvent evaporation while adhesion to metal is lost by hydrolytic degradation [8–10].

Adding nanofillers to a polymeric matrix can improve thermal and mechanical properties of nanocompounds [11]. Carbon nanofillers, including fullerene (0D) [12] and carbon nanotubes (CNT, 1D) [13], have been often used in polymers. Graphene, a 2D nanosheet, and a layer of sp2 hybridized carbon atoms are commonly used by researchers because of their mechanical, electric, thermal, optical, and barrier properties [14–18]. To take further

advantage of these properties, a graphene nanosheet (GNS) is reinforced in different polymeric matrices to produce nanocompounds. As a graphene derivative, graphene oxide (GO) maintains mechanical properties superior to those of graphene [19]. The presence of several functional groups containing oxygen on the GO surface constitutes an advantage to prevent sheet aggregation during the production of the compound material. Additionally, it can also provide more interaction sites on the interface between EP and nanofillers. The efficacy of GO reinforcement and hardening largely depends on the size [20], degree of exfoliation [21], dispersion state [22], and interface interactions between GO and the matrix [23].

Recently, several studies have proven that the ultrasonic treatment plays a key role in increased homogeneity of dispersion and size of GO sheets. This allows to obtain materials with better mechanical properties in biomedical and chemical engineering fields [24]. Ultrasonication promotes GO reduction through heat exchange between water microbubbles and GO sheets, with local pressure and temperature gradients that can exceed the critical point of water [25].

Among the covalent functionalization methods considered for GO dispersion in the polymeric matrix [26,27], the modification with GO silanes is a promising method to improve nanocompound properties. The alkoxy groups of the silane coupling agent and the GO hydroxyl groups participate in a chemical reaction. Then, the remaining functional groups of the silane molecules on GO sheets provide a chemical bond between the GO and the polymeric matrix [28]. On the other hand, organo-functional silanes have been directly used in epoxy nanocompounds to improve mechanical properties (Tensile) given the formation of -Si-O- stretching bonds within the organic epoxy chains. Therefore, it can be hypothesized that the performance of epoxy nanocompounds would be greatly improved using the advantages of silane-functionalized GO.

Pourhashem et al. [29] used GPTMS (3-glycidyloxypropyl trimethoxysilane) in GO functionalization. Their results indicate that compatibility between functionalized GO and the epoxy matrix improved. This promoted the storage module, glass transition temperature, thermal stability, traction and flexion properties, and fracture resistance of epoxy compounds. Furthermore, Wang et al. [30] used 3-aminopropyltriethoxysilane (APTES) for the covalent functionalization of graphene sheets. Their study proved that graphene sheets functionalized with silane agent are adequate for high interface interactions between graphene sheets and the epoxy matrix, even when there is a heavy nanofiller load (1% weight). So, thermal stability, resistance to traction, and strain-to-failure of the epoxy nanocompounds improved. Additionally, Li et al. [31] studied the mechanical properties of GO-epoxy nanocompounds functionalized with silane. They used two silane coupling agents with different terminals, including APTES and GPTMS. The nanocompounds containing aminofunctionalized GO (APTES-GO) showed a more significant increase in the Young's modulus and resistance to traction, while those containing GO functionalized with epoxy (GPTMS-GO) revealed a greater increase in ductility and fracture resistance. To obtain the nanocompounds, the researchers used three graphenes with different oxidation degrees and added GO without modifications at 1–3% weight. The same was done in samples with silane-functionalized GO (GO-GPTMS) at the same percentages. The chemical, structural, and morphological characteristics of the nanocompounds were analyzed. Furthermore, the dispersion quality of the nanocompounds in the polymeric matrix was assessed by ultrasonication, and the effect of GO at several oxidation degrees was observed on the mechanical properties of the nanocompounds obtained. Due to the above, this work contributes to the state-of-the-art by using an oxidized graphene system, graphene dispersion by ultrasonic process and chemical modification (GPTMS), creating a system that has not been studied and is not reported in the literature.

## 2. Materials and Methods

### 2.1. Materials

Graphenes with different oxidation degrees were provided by ID-nano (Cuauhtemoc, Mexico City, Mexico) CAS: 7782-42-5. Acetone (99.9% purity) CAS: 67-64-1 and 3-glycidyloxypropyl trimethoxysilane (GPTMS ≥ 98%) CAS: 2530-83-8 were obtained from Sigma-Aldrich (P.O. Box 14508, St. Louis, MO, USA). Toluene (99.9% purity) CAS: 108-88-3 was obtained from J.T. Baker (USA). Ethanol (99.9% purity) CAS: 64-17-5 was obtained from Merck (3050 Spruce St. Louis, MO, USA). Epoxy resin (DGEBA: Bisphenol A diglycidyl ether) CAS: 25068-38-6 and polyamide hardener (MP 100%) CAS: 68082-29-1 were obtained from DECOART, at a 2:1 ratio of resin:hardener.

### 2.2. Methods

#### 2.2.1. Pretreatment of GO

The GOs were placed in a vacuum pump at a 60 °C for 24 h. They were then placed in an oven at 100–120 °C for 24 h to eliminate traces of moisture. Finally, they were ground in a mortar and were sieved through a 200-sieve cloth according to ASTM E 11 to obtain powdered graphenes with an average particle size of 74 µm diameter.

#### 2.2.2. Functionalization of GO Using Silane (GPTMS)

The functionalization of GO using silane was carried out following the procedure in Figure 1. In a typical reaction, 150 mg GO was placed in a three-neck flask and 50 mL deionized water was added. The mix was dispersed by ultrasound in a water bath for 30 min, and 5 mL GPTMS was added in a constant drip. The mix was stirred at 30 °C for 3 h, 3 mL toluene was added to the solution, and the temperature was raised to 100 °C for 3 h. The functionalized GO was washed with ethanol trice and separated using a Teflon filter (0.45 µm) to eliminate the GPTMS molecule that did not react. Finally, the sample was dried in a vacuum oven at 80 °C for 24 h [32]. The functionalized GO material was called GO-GPTMS. Figure 2 shows the synthesis process of GO functionalized with GPTMS silane through the covalent interaction between GO and GPTMS. Toluene is used due to the polarity change that graphene undergoes when modified with GPTMS, thus allowing a better dispersion of the modified graphene.

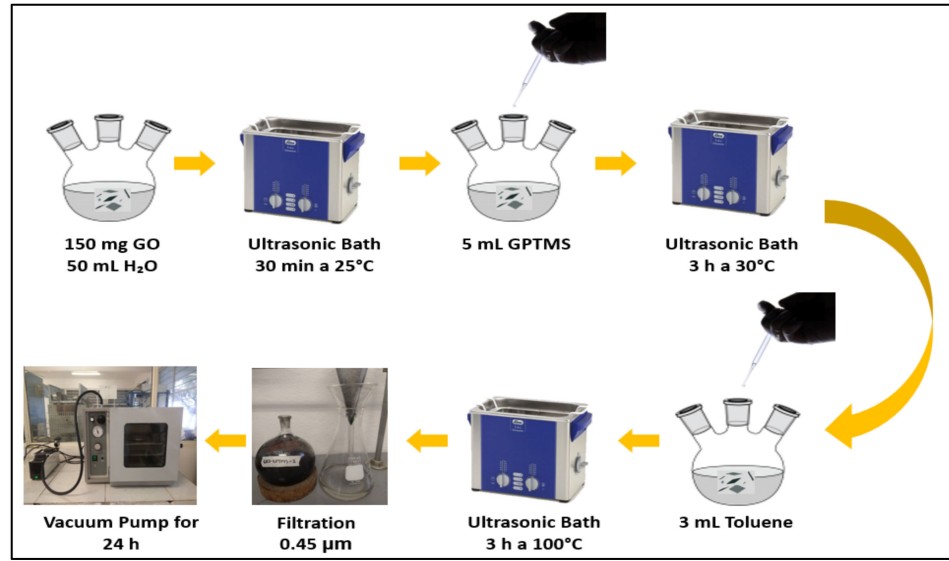

**Figure 1.** Functionalization process of GO with GPTMS.

#### 2.2.3. Preparation of GO/Epoxy and GO-GPTMS/Epoxy Nanocompounds

Table 1 shows the weight concentrations of oxidized graphenes (GO1, GO2, and GO3) as well as the concentration of the other elements that create the compound materials. The

GO was weighed and dispersed in acetone by ultrasonic agitation at room temperature for 1 h [33,34]. The EP was added to the solution and the mix was shaken in an ultrasonic bath for 30 min. The solution was then placed in a container for the evaporation of the solvent at room temperature for 24 h. Afterwards, the hardener was added at a 2:1 ratio (EP:hardener); it was manually shaken to obtain a homogenous phase for pouring in the respective mold (measuring cylinder, dog bone shape ASTM D-638). The molds were placed in a vacuum oven at room temperature for 15 min to eliminate the air bubbles created during the pouring process. The material was left to harden under environmental conditions for 24 h. It is important to have a control sample (target) of pure EP following the same process used to obtain EP/GO. Table 2 shows the working nomenclature matrix used during the production of nanocompounds.

**Figure 2.** Synthesis of functionalization process of GO with GPTMS.

**Table 1.** Weight concentration of GO and elements constituting the nanocomposites.

| Sample | %wt GO | Acetone | Epoxy Resin (EP) | Catalyst |
|--------|--------|---------|------------------|----------|
| 1 | 60 mg | 0.6 mL | 3.960 g | 1.980 g |
| 2 | 120 mg | 1.2 mL | 3.918 g | 1.962 g |
| 3 | 180 mg | 1.8 mL | 3.882 g | 1.938 g |

**Table 2.** Working nomenclature matrix.

| Material | Sample | |
|----------|--------|--|
| | **GO/Epoxy** | **GO-GPTMS/Epoxy** |
| EP | DGEBA GO1/Epoxy-1 | DGEBA GO1-GPTMS/Epoxy-1 |
| GO1 | GO1/Epoxy-2 GO1/Epoxy-3 | GO1-GPTMS/Epoxy-2 GO1-GPTMS/Epoxy-3 |
| GO2 | GO2/Epoxy-1 GO2/Epoxy-2 GO2/Epoxy-3 | GO2-GPTMS/Epoxy-1 GO2-GPTMS/Epoxy-2 GO2-GPTMS/Epoxy-3 |
| GO3 | GO3/Epoxy-1 GO3/Epoxy-2 GO3/Epoxy-3 | GO3-GPTMS/Epoxy-1 GO3-GPTMS/Epoxy-2 GO3-GPTMS/Epoxy-3 |

*2.3. Characterization*

2.3.1. Fourier Transform Infrared Spectroscopy

The Fourier transform infrared spectroscopy (FTIR) spectra were obtained in an FTIR spectrophotometer (Perkin Elmer's Spectrum Two, Seer Green, Beaconsfield H.) The wavelength range was 4000–650 cm$^{-1}$ in ATR mode, and 16 scans were performed.

2.3.2. Scanning Electron Microscopy (SEM) and Energy Dispersive X-ray Spectroscopy (EDS)

The morphology of GO powders and nanocomposites was examined by scanning electron microscopy (SEM) using a JEOL microscope (JSM 6010A, Akishima, Tokyo) in SEI mode at 1.5 kV. An EDS (energy dispersive X-ray spectroscopy) accessory was coupled to the SEM for evaluating the elemental analysis of GO powder and GO-GPTMS.

2.3.3. Optical Microscopy

An ABBE microscopies MPT01 optical microscope (Miamisburg, Ohio) was used to verify GO and GO-GPTMS dispersions in the EP matrix. A 100x optical lens was used.

2.3.4. Mechanical Tension Testing

Measuring cylinders conditioned according to ASTM D-68 (dog bone shape) were evaluated to assess the mechanical properties of the nanocompounds. An Instron 3340 (INSTRON, Norwood, Massachusetts) was used for tensile testing at a constant travel speed of 1 mm/min and a temperature of 25 °C. Young's modulus was measured using ASTMS E111-04 standard testing method for Young's modulus. Three samples per concentration were analyzed and mean values were calculated.

Young's modulus is calculated by means of load increment and the extension, between two points on the line as far apart as possible, where $Y$ is applied axial stress and $X$ strain data, using the following Equation (1):

$$E = \left( \frac{\sum XY - K\overline{XY}}{\sum X^2 - K\overline{X}^2} \right) \tag{1}$$

**3. Results**

This section may be divided by subheadings. It should provide a concise and precise description of the experimental results, their interpretation, as well as the experimental conclusions that can be drawn.

*3.1. Characterization of GO, GO-GPTMS, and Nanocompounds by FTIR Spectroscopy*

FTIR spectroscopy was used to characterize GO and study the silane graft in GO sheets (GO-GPTMS) after a drying process at 80 °C. Figure 3a shows the infrared spectrum of graphene with the highest degree of oxidation (GO1). In this image are the characteristic bands of the structure, along with hydroxyl, epoxy, carboxyl, and carbonyl functional groups in basal planes and borders, as described in the literature [35]. A wide band at 3400 cm$^{-1}$ corresponded to the stretching vibration of the hydroxyl-OH; asymmetrical and symmetrical stretching of -CH2 of the aliphatic section was observed at 2900 cm$^{-1}$; the band at 1600 cm$^{-1}$ corresponded to the vibration of C=C stretching (due to the conjugated structure of graphene oxide); the stretching vibration characteristic of carbonyl C=O groups bound to the rings of the structure and contained in carboxyls -COOH (carboxylic acid group) is observed at 1702 cm$^{-1}$; and the stretching vibration of C-OH is observed at 1150 cm$^{-1}$. Additionally, two bands are observed at 1035 and 870 cm$^{-1}$, attributing to the stretching vibrations of epoxy C-O-C and alkoxy C-O groups, respectively. These results show that GO1 has a structure with abundant oxygen-based functional groups. Figure 3b shows the infrared spectra of graphenes with the least degree of oxidation (GO2 and GO3). In them there is a marked reduction in the intensity of the bands associated to the oxygen-based functional groups as compared against the GO1 bands. That is, GO2

and GO3 contain a reduction that leads to the loss of oxygenated functional groups, as evidenced by the increase in sp2 hybridization bonds [36]. It must be considered that a reduction in the functional groups above due to a lower degree of oxygenation leads to an increase in the band characteristic of the structure, such as that at 1600 cm$^{-1}$ corresponding to the stretching vibrations of the C=C bond, while the one at 1160 cm$^{-1}$ corresponds to the stretching vibrations of the C-C bond, and that at 1430 cm$^{-1}$, to the vibrations characteristic of the scissor bending of -CH$_2$. Figure 3c shows the FTIR spectrum of silane-GPTMS. The characteristic bands of its structure are observed: two bands of the methyl group (-CH$_2$) with the epoxy-GPTMS group are found at 910 and 820 cm$^{-1}$, two stronger signals are observed at 1195 and 1090 cm$^{-1}$ corresponding to the vibrations of the siloxalkyl group (Si-O-CH$_3$) in GPTMS. Additionally, two vibrational bands at 2950 and 2850 cm$^{-1}$ are characteristic of the asymmetrical and symmetrical stretching of -CH, -CH$_2$, and CH$_3$. Similar spectral patterns have been observed by Li et al. [37].

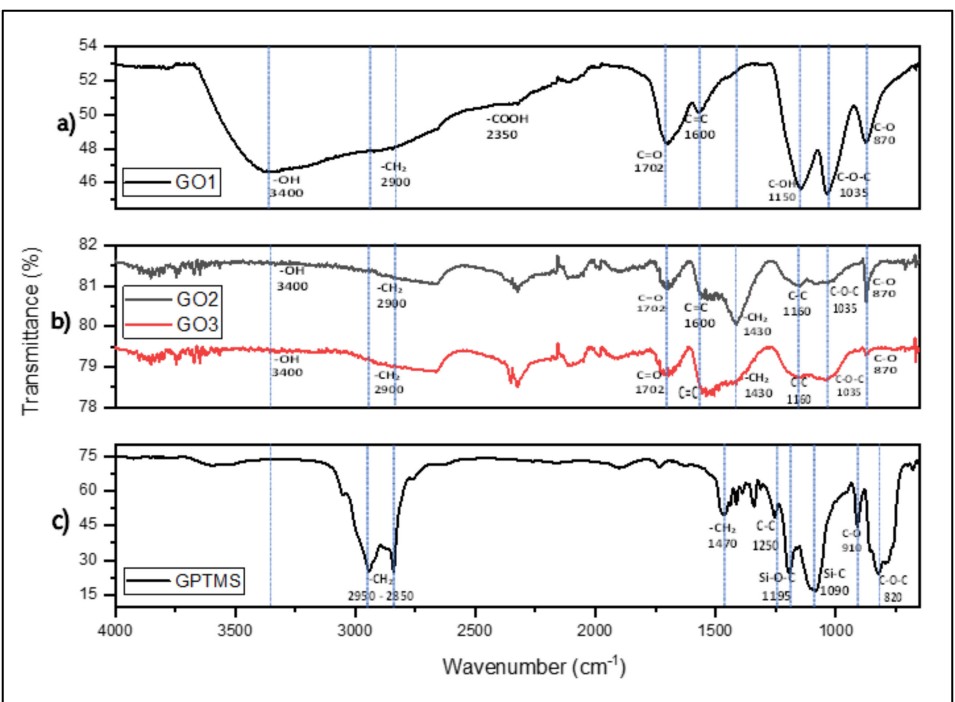

**Figure 3.** FTIR-ATR spectra of (**a**) GO1, (**b**) GO2, GO3, and (**c**) GPTMS.

Figure 4 shows the FTIR-ATR spectrum of the graphene silane-functionalized oxide (GO1-GPTMS, GO2-GPTMS, and GO3-GPTMS) samples after functionalization of GO with GPTMS. The band at 3400 cm$^{-1}$ was weakened, and two new bands appear at 2950 and 2870 cm$^{-1}$, corresponding to the stretching of the -CH2 groups of the alkyl chains assigned to silane. The stretching bonds of the Si@O-C and Si-O-Si groups were assigned at 1085 and 1000 cm$^{-1}$, respectively, indicating the chemical functionalization was successful. In addition, the characteristic signal at 900 cm$^{-1}$ confirms the presence of a large number of epoxy groups in the GO modified with GPTMS. The signals of previous FTIR suggest that the -OH groups of the GO sheets might only react with the -OCH3 groups of GPTMS molecules. Given that the peak characteristic of GO, including the C=O stretching of carboxylic (1720 cm$^{-1}$) and CH2 (2950 cm$^{-1}$) groups disappear or are smaller in the GO-GPTMS sample [38].

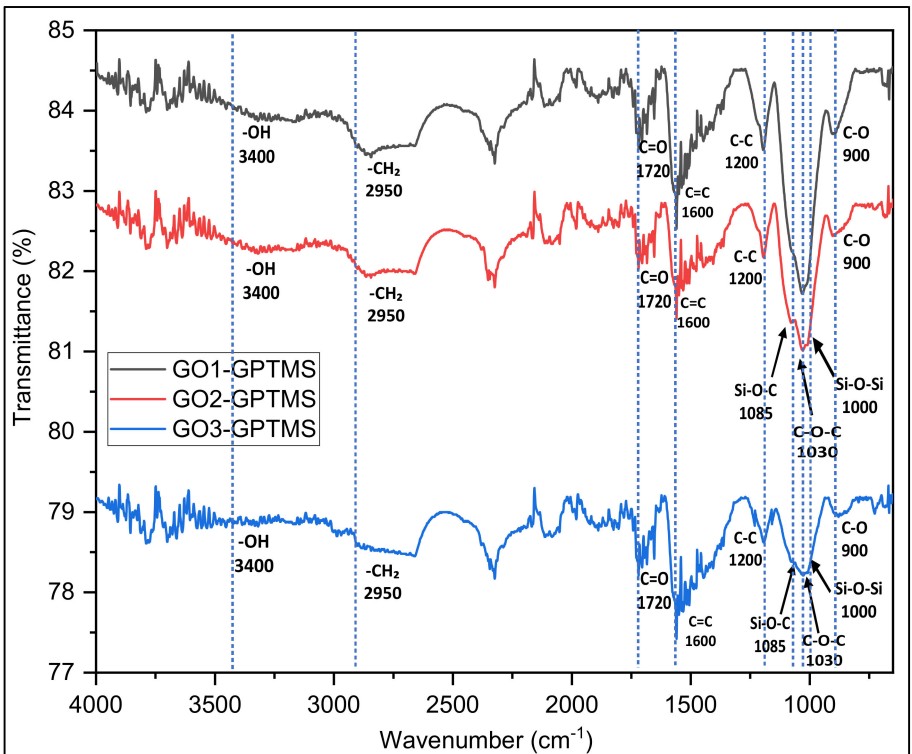

**Figure 4.** FTIR spectra of GOs functionalized with silane GPTMS.

A quantitative analysis of the areas under the curve of some relevant signals of the FTIR spectra was carried out following the procedure described by Guerrero et al. [36]:

1. The bands were measured using originPro, (integration option) to obtain the areas of each selected peak or signal.
2. A polynomial baseline was drawn from the raw spectra.
3. The resulting spectrum was multiplied by −1, and the origin was set to y = 0 to obtain positive bands.

Table 3 shows the results of the areas of the bands in the FTIR-ATR spectra of the GOs used. The area of GO1 is the largest when compared against those of GO2 and GO3, which contain a marked decrease in the areas of the bands linked to oxygen-based functional groups. There is an increase in $sp^2$ hybridization bonds. The results are characteristic of GOs since they contain different degrees of oxidation.

**Table 3.** Area under the curve of GO signals.

| GOs | -OH | -CH₂ | -COOH | C=O | C=C | C-OH | C-CH₂ | C-C | C-O-C | C-O |
|-----|-----|------|-------|-----|-----|------|-------|-----|-------|-----|
| GO1 | 84,319.7 | 5128.66 | 4912.41 | 13,917.2 | 12,355.6 | 9571.69 | - | - | 7374.74 | 11,531.9 |
| GO2 | 40,603.72 | 9028.46 | - | 11,700.27 | 10,756.05 | - | 15,674.61 | 12,592.19 | 19,341.69 | 6832.46 |
| GO3 | 31,760.92 | 14,156.32 | - | 11,304.44 | 11,802.26 | - | 16,037.42 | 12,706.46 | 16,115.51 | 4287.52 |

Table 4 presents the results of the areas of the bands corresponding to the FTIR-ATR spectra of GOs functionalized with silane. GO1-GPTMS contains the largest area of the bands of functional groups characteristic of its structure. When compared to GO3-GPTMS, its area was smaller since it depends on the degree of oxidation.

**Table 4.** Area under the curve of the signal corresponding to GOs functionalized with silane.

| Sample | -OH | -CH$_2$ | C=O | C=C | C-C | Si-O-C | C-O-C | Si-O-Si | C-O |
|---|---|---|---|---|---|---|---|---|---|
| GO1-GPTMS | 45,787.94 | 12,791.81 | 13,293.09 | 24,618.71 | 10,334.75 | 7311.22 | 4338.98 | 6312.62 | 10,332.37 |
| GO2-GPTMS | 44,145.55 | 11,314.32 | 11,794.27 | 25,673.36 | 9081.82 | 7038.82 | 4217.75 | 5586.69 | 9992.90 |
| GO3-GPTMS | 30,844.95 | 10,996.61 | 7151.31 | 25,175.67 | 8201.3 | 6050.94 | 3834.42 | 4962.36 | 8772.26 |

Figure 5A shows the FTIR-ATR spectrum of DGEBA S/E EP (without hardener) and DGEBA C/E (with hardener) when obtaining dry paint. Here, we present the signals characteristic of both resins. The vibrational mode of EP of R-O-R groups corresponding to the tension of the aromatic ether bond are found at 1234 and 1030 cm$^{-1}$ while the stretching of the terminal epoxy group C-O-C is observed at 820 cm$^{-1}$. The most characteristic vibrational bands for the hardener were those of amine and amide groups. For instance, the vibration of the characteristic stretching of C=O carbonyl groups of the amide function is observed at 1600 cm$^{-1}$, while that of C-N, N-H bands of amide II are found at 1127 cm$^{-1}$ and that of amide I, at 3200 cm$^{-1}$ [39]. Figure 5B shows the GO2/epoxy nanocompounds at concentrations of 1%, 2%, and 3% of GO load where the band at 3400 cm$^{-1}$ corresponds to the stretching vibration of hydroxyl -OH. At 3200 cm$^{-1}$, there is the characteristic band of amide N-H, and the asymmetrical and symmetrical stretching of -CH$_3$, -CH$_2$, and -CH is found at 2960, 2920, and 2870 cm$^{-1}$. The stretching vibration characteristic of carbonyl groups C=O is observed at 1740 cm$^{-1}$; double bond C=C stretching is at 1600 cm$^{-1}$; vibrations of the C-C bond stretching are at 1200 cm$^{-1}$; the vibration characteristic of the scissor bending of -CH$_2$ is at 1509 cm$^{-1}$; and 720 cm$^{-1}$ corresponds to the bending of the alternate movement of -CH$_2$. Additionally, there are two bands at 1100 cm$^{-1}$ and 820 cm$^{-1}$ that can be attributed to the stretching vibrations of C-O-C epoxy and C-O alkoxy groups, respectively. Figure 5C shows the FTIR spectra corresponding to the GO2-GPTMS/epoxy nanocompounds at concentrations of 1%, 2%, and 3%. These FTIR correspond to the nanocompounds made with silanized graphene oxide, proving that functionalization was carried out successfully. There are two characteristic signals of the Si-O-Si and Si-O-C bonds between 1030 and 1230 cm$^{-1}$, corresponding to the silane used. There is also a band at 3400 cm$^{-1}$ that corresponds to the stretching vibration of -OH hydroxyl, while a band characteristic of N-H amide is observed at 3200 cm$^{-1}$. Asymmetrical and symmetrical stretching of -CH$_3$, -CH$_2$, and -CH was observed at 2960, 2920, and 2870 cm$^{-1}$. The stretching vibration characteristic of the C=O carbonyl groups is observed at 1740 cm$^{-1}$; double bond C=C stretching is observed at 1600 cm$^{-1}$; stretching vibrations of C-C bond at 1200 cm$^{-1}$; the vibration characteristic of scissor bending of -CH$_2$ at 1509 cm$^{-1}$; and bending vibrations of the alternate movement of -CH$_2$ at 720 cm$^{-1}$. Additionally, two bands are observed at 1100 and 820 cm$^{-1}$, which can be attributed to the stretching vibrations of epoxy C-O-C and alkoxy C-O groups.

*3.2. Morphological Characterization*

Graphene oxides were morphologically characterized to observe the surface they presented. Figure 6 shows the SEM images of the three different graphene oxides without functionalization with silane and functionalized with GPTMS.

3.2.1. Characterization of Silane-GPTMS Functionalized GOs and GOs by Scanning Electron Microscopy (SEM/EDS)

Figure 6A–C present the morphology of a rougher surface and thin sheets with swollen structure for GO1, GO2, GO3 respectively. Figure 6D–F show a smoother morphological surface with a layered structure on the borders of GO1-GPTMS, GO2-GPTMS, and GO3-GPTMS respectively. Pourhashem et al. (2017) [40] consider that silane agent acts as a buffer between GO sheets and prevents the stacking of GO sheets. The differences observed in the morphology of samples in Figure 6, GOs and GOs-GPTMS, are the result of a silane

covalent bond in GO and the condensation reactions between silane molecules in GO sheets, that hamper GO agglomeration, as described by Kim et al. [41].

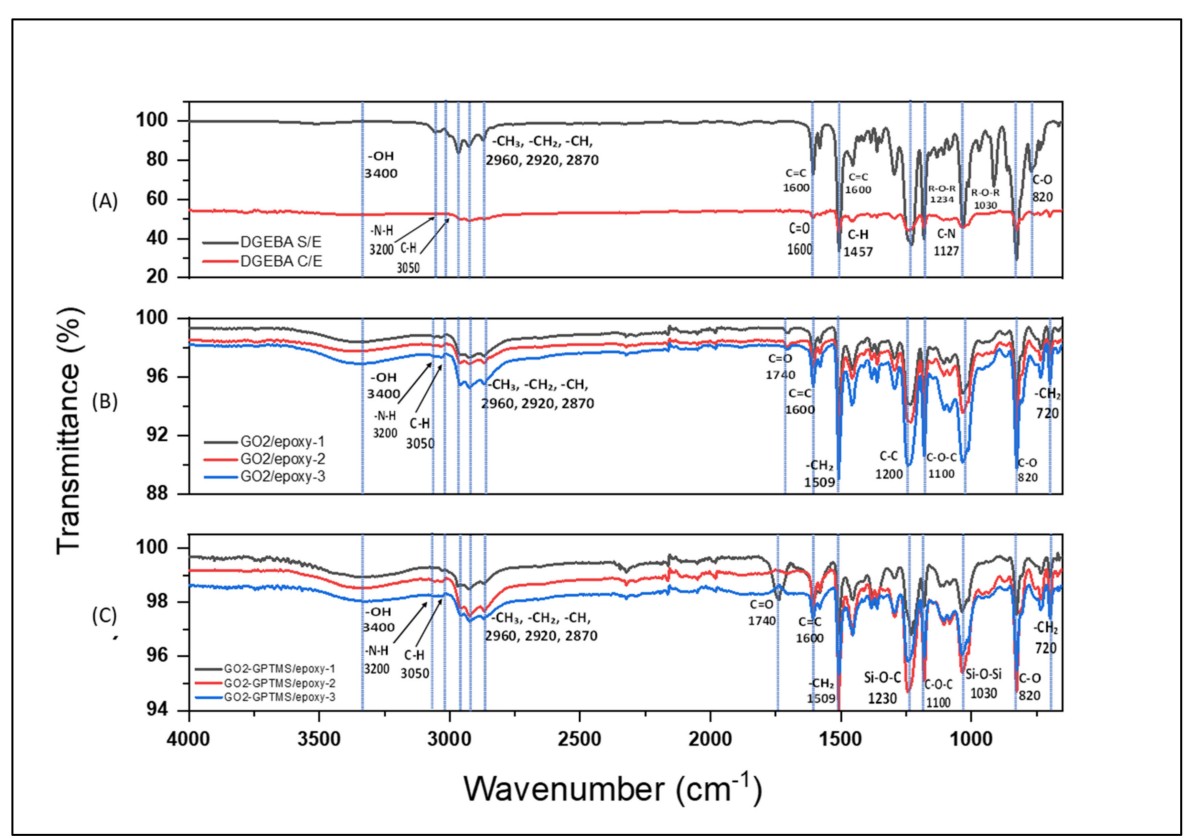

**Figure 5.** FTIR spectrum (**A**) DGEBA S/E and DGEBA C/E epoxy resin, (**B**) GO2/epoxy nanocompounds, and (**C**) GO2-GPTMS/epoxy nanocompounds.

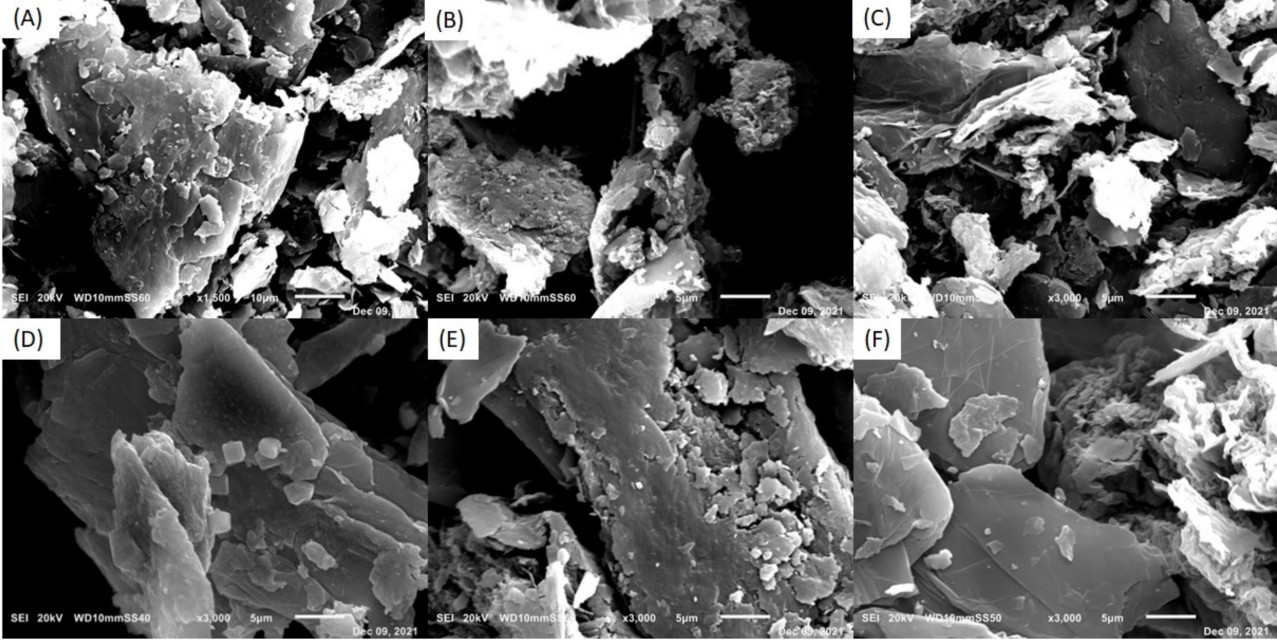

**Figure 6.** SEM micrographs (3000x) of graphene oxides: (**A**) GO1, (**B**) GO2, (**C**) GO3, (**D**) GO1-GPTMS, (**E**) GO2-GPTMS, and (**F**) GO3-GPTMS.

In graphenes, with different oxidation degrees, the characteristic elements of the formulation is present (hydroxyl, epoxy, carboxyl, and carbonyl functional groups) in basal planes and borders [42,43]. On the other hand, functionalized graphenes show the presence of silicon due to the silane agent (GPTMS) used in the functionalization process. The epoxy groups of GPTMS are reactive and easily hydrolyzed to create silanol groups (Si-OH) [44]. The GO hydroxyl groups would react with GPTMS hydrolyzed ethoxy groups, leading to GPTMS graft in GO sheets through Si-O-C bonds [45]. Additionally, free silanol groups (Si-OH) would react between them to create Si-O-Si in GO sheets (see Figure 7) [46].

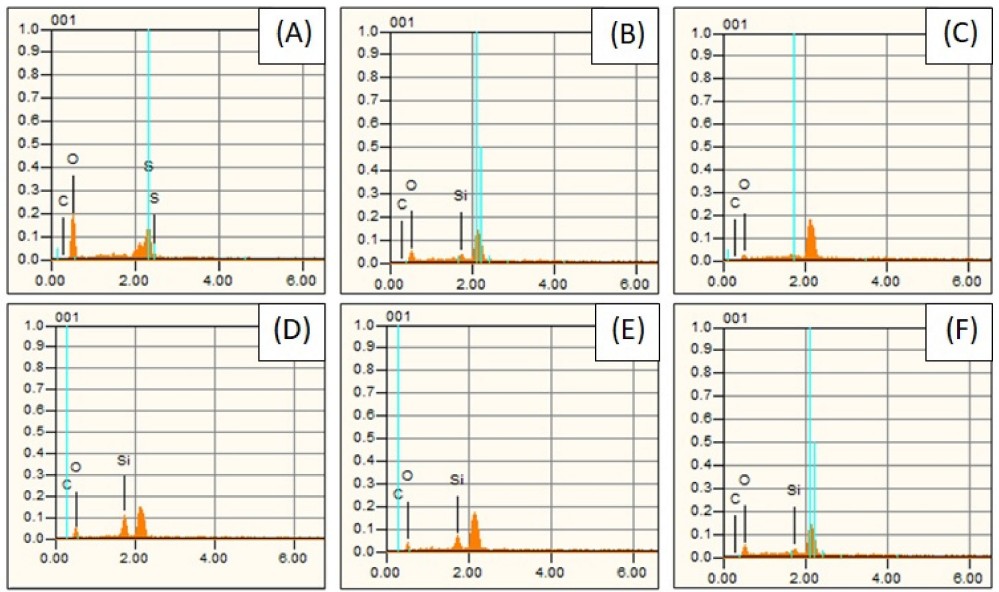

**Figure 7.** EDS analysis of graphene oxides (**A**) GO1, (**B**) GO2, (**C**) GO3, (**D**) GO1-GPTMS, (**E**) GO2-GPTMS, and (**F**) GO3-GPTMS.

### 3.2.2. Interface and Microstructure of Nanocompounds

The fracture surface of the samples was observed after the traction assay to explore the interfacial quality of the composites. The improvement in the properties of the epoxy matrix is possible owing to the homogeneous dispersion of the filler in the matrix and the strong interface between the matrix and the filler. To understand the influence of graphene oxides on the properties of epoxy compounds, the fracture surfaces of the samples were characterized through SEM, as shown in Figure 8A,B at two different magnifications. The fracture surface of pure epoxy is quite smooth and shows some crevices, a typical characteristic of fragility fractures [47].

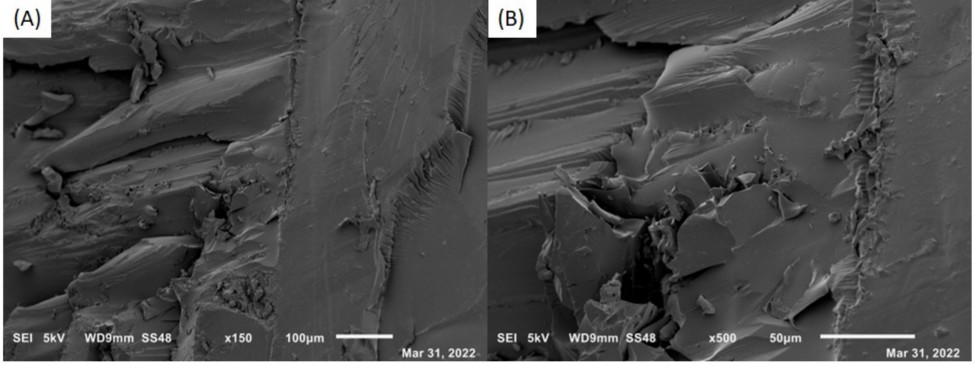

**Figure 8.** SEM micrographs (**A**) (150×) and (**B**) (500×) of pure DGEBA epoxy resin.

To understand GO dispersion in DGEBA epoxy resin, Figure 9 shows SEM micrographs of GO2/epoxy-1, GO2/epoxy-2, and GO2/epoxy-3 nanocompounds. There is an increase in the roughness of the material and GO is evidently present in most of the surface, while GO2/epoxy-3 shows a greater presence of GO in the transverse section as compared against GO2/epoxy-1 and GO2/epoxy-2. This is explained by the increase in GO concentration. Additionally, there are some dimple voids on the surface (see Figure 9A–C). GO aggregates can be seen in the center of the dimples, as indicated by the white circles. During the failure process, these aggregates can induce microcracks, evident voids between the matrix and the aggregates (Figure 9D–F), indicating a relatively weak interfacial bonding between some GO sheets and EP.

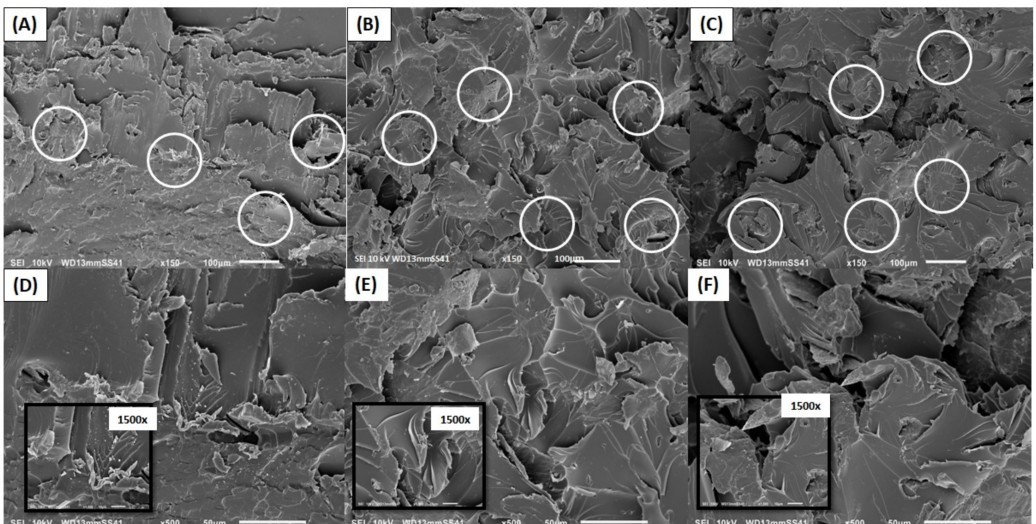

**Figure 9.** SEM micrographs (150×), (500×), and (1500×); (**A,D**) GO2/epoxy-1, (**B,E**) GO2/epoxy-2, and (**C,F**) GO2/epoxy-3.

Figure 10 shows the SEM micrographs of GO2-GPTMS/epoxy-1, GO2-GPTMS/epoxy-2, and GO2-GPTMS/epoxy-3 nanocompounds that present similar rough surfaces. There are no evident GO-GPTMS/epoxy sheet groupings on the fracture surface. The SEM image of high magnification in Figure 10A reveals a good relative dispersion of GO-GPTMS/epoxy. Additionally, the interface bonding between the GO-GPTMS sheet and the epoxy matrix improves after the superficial silane functionalization given that no voids are observed between the GO sheet and the matrix on the fracture surface. A careful observation suggests that some EP molecules seem to be grafted into the GO-GPTMS/epoxy surface (see Figure 11), which differs from the relatively smooth and clear surface of the GO sheet shown in Figure 10. This phenomenon would promote the transference of local tension from the matrix to the sheet efficiently through the improved interface matrix/GO sheet [48].

### 3.3. Characterization of GO/Epoxy and GO-GPTMS/Epoxy Nanocompounds by Optical Microscopy

Optical microscopy (light transmitted) was used to establish an initial approach and identify the morphology of the nanocompounds due to the transparency of EP. Figure 12 shows a comparison between the optical micrographs at 100× of the nanocompounds prepared at different times of GO dispersion in acetone.

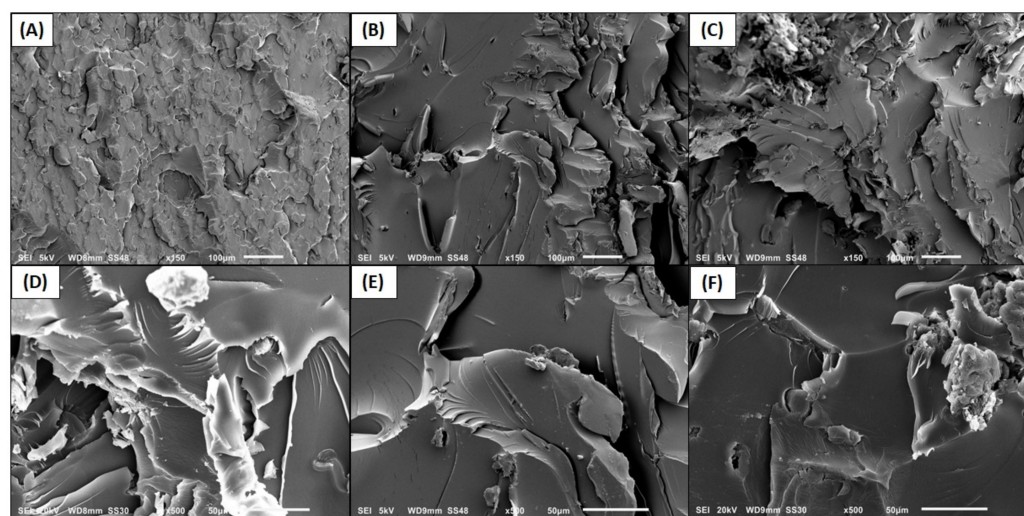

**Figure 10.** SEM micrographs (150×), (500×); (**A**,**D**) GO2-GPTMS/epoxy-1, (**B**,**E**) GO2-GPTMS/epoxy-2, and (**C**,**F**) GO2-GPTMS/epoxy-3.

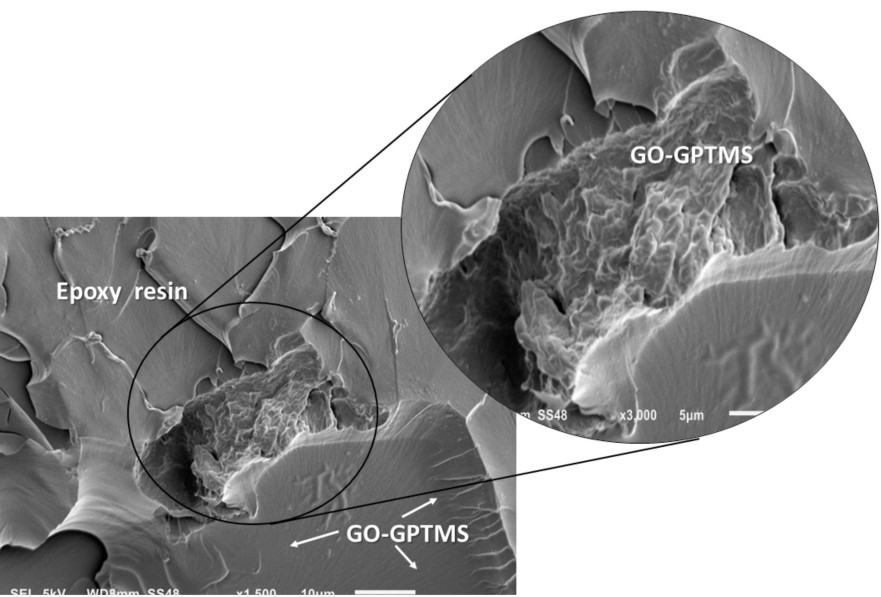

**Figure 11.** SEM micrographs (1500×), (3000×) of GO2-GPTMS-1 nanocomposite.

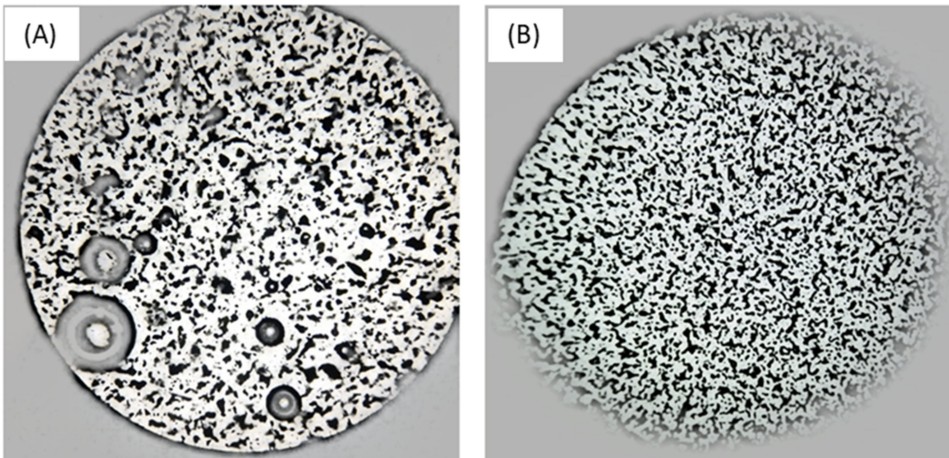

**Figure 12.** Micrographs of nanocompounds (100×) where GO dispersion in acetone was carried out at different times: (**A**) 30 min and (**B**) 60 min.

The mixing method by ultrasonic shaking helped to observe GO dispersion in acetone at two different times. Figure 12A shows the appearance of agglomerates at 30 min while Figure 12B presents a good dispersion of GO particles in the polymeric matrix at 60 min.

### 3.4. Mechanical Characterization of Nanocompounds

The two types of modified GO were incorporated to the EP and the mechanical properties of GO/epoxy and GO-GPTMS/epoxy nanocompounds were characterized. It was proven that all the compound samples containing GPTMS-functionalized GO showed a significant improvement in mechanical properties against pure EP [49]. In Figure 13A, the samples with an increase in their mechanical properties under tension are shown. The values of Young's modulus and resistance to traction obtained from tension testing are shown in Figures 13B and 14. The figures show that the samples with compound GO-GPTMS/epoxy material (1% weight GO) exhibited values higher in Young's modulus (2.096 GPa) and resistance to traction (98.713 MPa), around 35% and 52% higher than in pure epoxy. An additional increase in GO load leads to the degradation of the traction properties. However, a load above 1% weight GO shows differences between Young's modulus and a minor resistance to traction between GO/epoxy (3% weight GO) and GO-GPTMS/epoxy (3% weight GO).

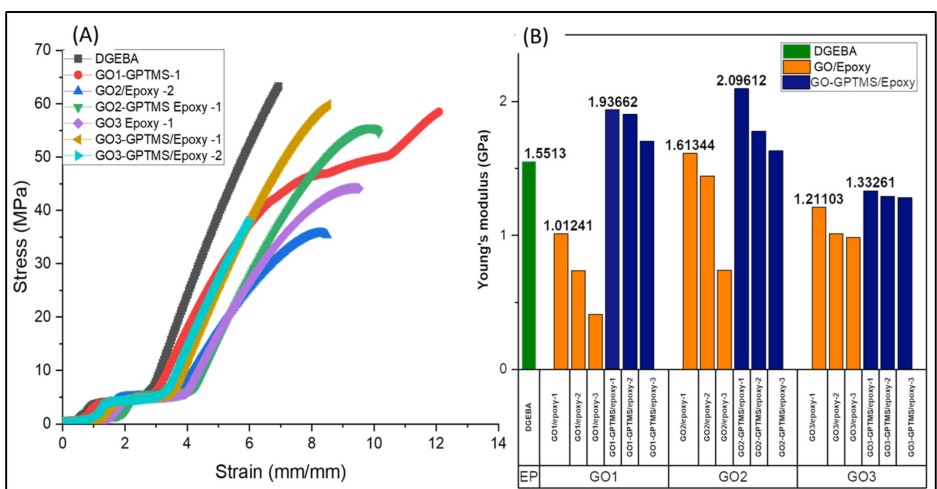

**Figure 13.** (**A**) Stress strain diagram for nanocompound. (**B**) Young's modulus of DGEBA, GO/epoxy, and GO-GPTMS/epoxy nanocomposites.

Figure 14 shows the average stresses measured in the nanocomposites, which is shown in GO1-GPTMS/Epoxy-1; the G01 group being the one that increased the response not only in stresses but also with 13,195 deformation and the GO2-GPTMS/Epoxy-1 sample showed an increase of 98.713 MPa compared to the DGEBA with 64.778 ± 2.02, both being those with 1% by weight that are shown as evidence of the increase in mechanical properties for these samples in the graph.

Ductility was also improved with the increase in peak deformation regarding traction by 11 and 51% in materials created with 1% load GO in GO/epoxy and GO-GPTMS/epoxy, respectively (see Table 5). Unlike the results of resistance to traction and Young's modulus, the improvement in peak deformation is more evident in nanocomposites containing GO-GPTMS/epoxy [46,47].

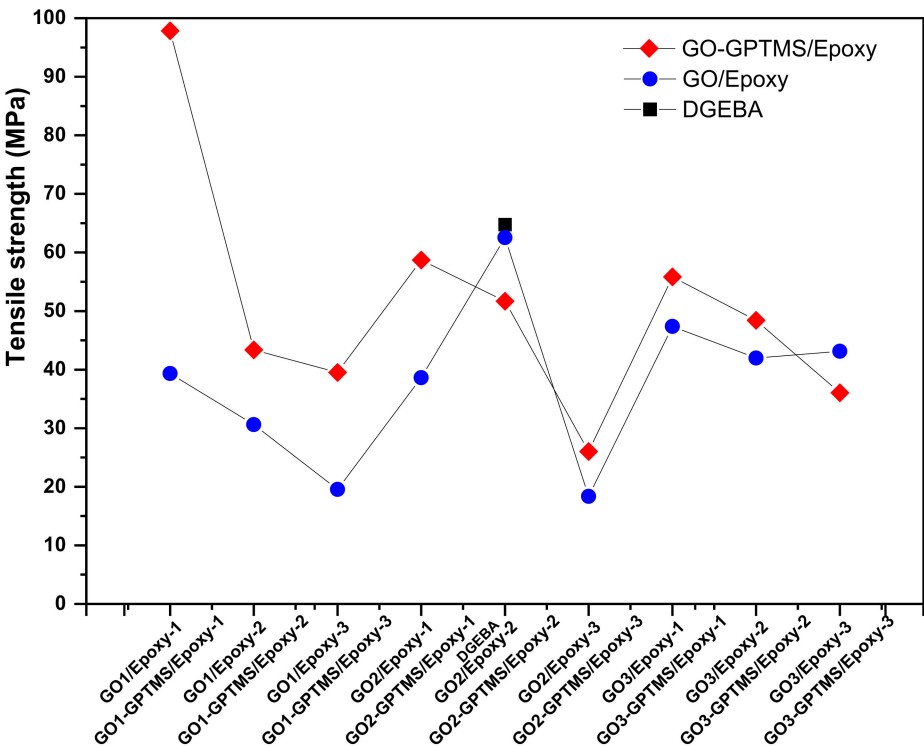

**Figure 14.** Resistance to traction of DGEBA, GO/epoxy, and GO-GPTMS/epoxy nanocomposites.

**Table 5.** Mechanical properties of pure epoxy and compound materials GO/epoxy and GO-GPTMS/epoxy between 1 and 3% weight GO.

| Sample | Stress (MPa) | Young Module (GPa) | Strain Max (%) |
|---|---|---|---|
| DGEBA | 64.778 ± 2.02 | 1.551 ± 0.52 | 9.860 ± 0.72 |
| GO1/Epoxy-1 | 39.332 ± 1.33 | 1.012 ± 0.46 | 10.573 ± 0.39 |
| GO1/Epoxy-2 | 30.603 ± 1.35 | 0.736 ± 0.18 | 10.340 ± 0.16 |
| GO1/Epoxy-3 | 19.550 ± 1.35 | 0.409 ± 0.33 | 10.870 ± 0.78 |
| GO1-GPTMS/Epoxy-1 | 87.836 ± 1.17 | 1.936 ± 0.27 | 13.195 ± 0.89 |
| GO1-GPTMS/Epoxy-2 | 63.358 ± 1.61 | 1.905 ± 0.51 | 10.415 ± 0.65 |
| GO1-GPTMS/Epoxy-3 | 59.524 ± 1.36 | 1.704 ± 0.92 | 10.567 ± 0.42 |
| GO2/Epoxy-1 | 38.623 ± 1.78 | 1.613 ± 0.63 | 10.917 ± 0.83 |
| GO2/Epoxy-2 | 62.528 ± 1.53 | 1.443 ± 0.32 | 9.763 ± 0.52 |
| GO2/Epoxy-3 | 18.357 ± 1.23 | 0.740 ± 0.05 | 9.041 ± 0.20 |
| GO2-GPTMS/Epoxy-1 | 98.713 ± 1.23 | 2.096 ± 0.19 | 14.90 ± 0.19 |
| GO2-GPTMS/Epoxy-2 | 51.688 ± 1.25 | 1.777 ± 0.33 | 10.530 ± 0.51 |
| GO2-GPTMS/Epoxy-3 | 46.017 ± 1.78 | 1.631 ± 0.33 | 10.899 ± 0.97 |
| GO3/Epoxy-1 | 47.363 ± 2.65 | 1.211 ± 0.61 | 9.381 ± 0.42 |
| GO3/Epoxy-2 | 41.970 ± 2.03 | 1.010 ± 0.39 | 8.353 ± 0.67 |
| GO3/Epoxy-3 | 33.145 ± 2.85 | 0.984 ± 0.19 | 8.138 ± 0.45 |
| GO3-GPTMS/Epoxy-1 | 55.836 ± 1.12 | 1.332 ± 0.52 | 9.446 ± 0.66 |
| GO3-GPTMS/Epoxy-2 | 52.417 ± 1.79 | 1.291 ± 0.17 | 9.765 ± 0.35 |
| GO3-GPTMS/Epoxy-3 | 46.038 ± 1.45 | 1.281 ± 0.49 | 9.736 ± 0.29 |

The improved dispersion and interfacial bonding in the composites is likely due to the compatibility and covalent reaction between the epoxy matrix and the GPTMS chains grafted on the GO surface. Figure 15 contains the molecular structure of the hardeners and DGEBA resin, showing that the latter presents epoxy groups. The GPTMS chain in the GO-GPTMS contains terminal epoxide groups as those of EP (blue dotted circle) that can improve compatibility and miscibility between GO-GPTMS and the epoxy matrix. On the other hand, epoxy functional groups introduced from GPTMS into GO-GPTMS sheets can polymerize with EP during hardening to create epoxy compounds. The proposed

formation mechanism of the compounds is presented in Figure 15. The functionalized epoxy groups on the surface of GO sheets react with EP monomers and the sheets can strongly bind to the epoxy matrix through the covalent bridge [48,49].

**Figure 15.** Schematic illustration of the GO-GPTMS/epoxy system.

## 4. Conclusions

This work presents ultrasound as an ecological and affordable method of chemical synthesis to prepare epoxy nanocomposites with GO-SFS and GO-GPTMS at different loads (1%, 2%, and 3% weight GO). We investigated and compared the effect of silane functionalization of GO sheets and their load on mechanical properties. The FTIR-ATR results proved that the GPTMS coupling agent containing epoxy groups was successfully grafted on the surface of GO sheets. The SEM analyses revealed that the covalent functionalization of silane with GO produced a better dispersion of GO-GPTMS and a strong interface interaction with the epoxy matrix. This was reflected in GO2-GPTMS/epoxy and the 1% concentration in weight GO. An increase in Young's modulus was also observed along with a higher resistance to traction in nanocomposites filled with GO-GPTMS/epoxy as compared against the compound materials of pure epoxy and GO-SFS/epoxy.

**Author Contributions:** Conceptualization, R.S.-D. and A.M.S.-D.; methodology, R.H.-P.; software, A.O.-P. and E.G.G.-M.; validation, A.O.-P., E.G.G.-M. and A.M.S.-D.; investigation, A.O.-P. and E.G.G.-M.; resources, A.O.-P. and J.A.S.-C.; data curation, E.G.G.-M.; writing—original draft preparation, A.O.-P.; writing—review and editing, A.O.-P. and R.S.-D.; visualization, J.A.S.-C. and E.G.G.-M.; supervision, A.O.-P.; project administration, A.O.-P.; funding acquisition, Tecnológico Nacional de Mexico and Consejo Nacional de Ciencia y Tecnología. All authors have read and agreed to the published version of the manuscript.

**Funding:** Tecnológico Nacional de Mexico and Consejo Nacional de Ciencia y Tecnología for the facilities granted in the use of the facilities and the scholarship of the student González-Mondragón, E.G respectively.

**Informed Consent Statement:** Not applicable.

**Data Availability Statement:** We have presented all the data in the article, now we do not have a website to store our data, they can clarify any doubt with the corresponding author.

**Conflicts of Interest:** The authors declare no conflict of interest.

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
