# Peer review of "Obtention and Characterization of GO/Epoxy and GO-GPTMS/Epoxy Nanocompounds with Different Oxidation Degrees and Ultrasound Methods"

_carbon, 2022_

Round 1

Reviewer 1 Report

Dear Editor and Authors,

the paper is interesting but some issues must be solved.

1.     Present in the Introduction part, the novelty of the study.

2.     Explain why you added toluene to the mixture, while the synthesis of modified GO.

3.     Figure 2 is not decided. The epoxy ring doesn’t react with OH from graphene oxide?

4.     The Raman characterization must be added to the materials obtained.

5.     In Figure 15, the epoxy ring reacts with another epoxy??? Please add some references

6.     The English must be checked.

Author Response

Present in the Introduction part, the novelty of the study.

  • Due to the above, this work contributes to the state of the art by using an oxidized graphene system, graphene dispersion by ultrasonic process and chemical modification (GPTMS), creating a system that has not been studied and is not reported in the literature.

Explain why you added toluene to the mixture, while the synthesis of modified GO.

  • Toluene is used due to the polarity change that graphene undergoes when modified with GPTMS, thus allowing a better dispersion of the modified graphene.

Figure 2 is not decided. The epoxy ring doesn’t react with OH from graphene oxide?

  • Figure 2 has been corrected

The Raman characterization must be added to the materials obtained.

  • The analysis of functional groups was carried out using FTIR-ATR but Raman is not used because

the molecules that are used and obtained have a dipole moment.

In Figure 15, the epoxy ring reacts with another epoxy??? Please add some references

  • Figure 15 has been corrected and the references are included.

The English must be checked.

Reviewer 2 Report

The results of the spectral analysis seem to me to be not explained in proper way. It would be nice is the authors explain the relation of spectral characteristics of composite with the mechanical characteristics.

The following sections are present more traditional microscopic analysis of composite.

It seems the authors try to connect the molecular (the "chemical") approach and the micromechanical approach. This is a worthy goal, which seems to be not achieved.

If consider the paper as technical report, it must be estimated as a qualified technical work, which may be of interest to specialists in the field of epoxy resin - graphene composites and it may be published.

Author Response

The results of the spectral analysis seem to me to be not explained in proper way. It would be nice is the authors explain the relation of spectral characteristics of composite with the mechanical characteristics.

  • I detailed the results of the SEM images in the manuscript.
  • We conducted molecular studies with mechanical properties, relating them to the impact they had as a whole

Reviewer 3 Report

In this article, the authors used ultrasound to prepare epoxy nanocomposites with GP-SFS and GO-GPTMS successfully. They found that silane functionalization of GO sheets helped the improvement in mechanical properties of nanocompounds because GPTMS coupling agent containing epoxy groups was grafted on the surface of GO sheets. I enjoyed reading the manuscript and recommend it to be published in Journal of Carbon Research. However, prior to acceptance, authors should address the following minor comments:

1. In Introduction, “As a graphene derivative, graphene oxide (GO) maintains mechanical properties superior to those of graphene.” However, as is known, Young’s modulus and strength for graphene is higher than GO. Please be more specific what kinds of mechanical properties are referred to here.

2. In 2.2.1, “To obtain powdered graphenes with an average particle size and 74 um diameter measured.” This sentence is not completed.

3. In 2.2.2, “the functionalized GO was washed…” should be “The functionalized GO was washed…”.

4. In Table 2, what is DGEBA? Full name cannot be found. In addition, the author should add lines in the table to divide EP, GO1, GO2 and GO3. And the nomenclature should be clear, like GO1/Epoxy-1, the first 1 is about the degree of oxidation and the second 1 is about the GO concentration. The information about how to determine the different degree of oxidation is also needed.

5. The title for 2.3.2 should be “Scanning electron microscopy (SEM) and energy dispersive X-ray spectroscopy (EDS)”

6. In 3, all subtitle should be 3. Not 4..

7. In 3, -CH2 should be -CH2.

8. In Figure 3, why some blue lines are not at the peak position?

9. In 3.1, “The stretching bonds of the Si-O-C and Si-O-Si groups were assigned at 1083 and 1047 cm-1”. These numbers are different from those shown in Figure 4.

10. In Table 3, what do the orange color mean?

11. In 3.1, “…at 1600 cm-1 is the double bond C=C stretching, at 12 cm-1 are the stretching vibrations of C-C bond…” Where is the number 12 from?

12. In 3.2, “Figure 5 shows the SEM images of the three different graphene oxides…” should be “Figure 6 shows the SEM images of the three different graphene oxides…”.

13. The format for figure number is not consistent, like Figure 6[a), Figure 8 (a and b), Figure 9 a, b…

14. In 3.2.2, “This phenomenon, would promote the transference o…” What is o?

15. In Figure 11, “Resina epoxy” should be “Resin epoxy”.

16. In 3.4, please add details about force-displacement curves and how authors calculated Young’s modulus and resistance to traction.

17. In 3.4, “…exhibited values higher than…” should be “…exhibited values higher…”.

18. In 3.4, please provide evidences for the statement “However, a load above 1% weight GO evidences differences between Young’s modulus and a minor resistance to traction between GO/epoxy (3% weight GO) and GO-GPTMS/epoxy (3% weight GO).

19. In 3.4, “…with the increase in peak deformation regarding traction by 11 and 51%...” How did authors calculate this? The degree of oxidation was not mentioned.

20. In 3.4, “Unlike the results of resistance to traction and Young’s modulus…”. I don’t think this statement is correct because Table 5 showed improvement in all resistance to traction, Young’s modulus and peak deformation in nanocomposites containing GO-GPTMS/epoxy.

21. In Figure 14, curves are usually to show the changing trend, therefore, curves figure shouldn’t be used here.

22. In 3.4, the blue dotted circle cannot be seen.

Author Response

In Introduction, “As a graphene derivative, graphene oxide (GO) maintains mechanical properties superior to those of graphene.” However, as is known, Young’s modulus and strength for graphene is higher than GO. Please be more specific what kinds of mechanical properties are referred to here.

The word “Tensile” is included.

In 2.2.1, “To obtain powdered graphenes with an average particle size and 74 um diameter measured.” This sentence is not completed.

the word "and" is changed and replaced by "of"

In 2.2.2, “the functionalized GO was washed…” should be “The functionalized GO was washed…”.

The word "the" was changed to "The"

In Table 2, what is DGEBA? Full name cannot be found. In addition, the author should add lines in the table to divide EP, GO1, GO2 and GO3. And the nomenclature should be clear, like GO1/Epoxy-1, the first 1 is about the degree of oxidation and the second 1 is about the GO concentration. The information about how to determine the different degree of oxidation is also needed.

The word DGEBA is included in the reagents. Oxidized graphene and two graphenes with degrees of reduction were used. Reduced graphenes are considered graphenes with low degrees of oxidation. The way to characterize these last graphenes was through FTIR analysis where a decrease in OH and carbonyl functional groups is observed due to reduction. In this way, the different degrees of graphene oxidation are worked on. In addition, the technical sheet provided by the supplier does not specify the degrees of oxidation.

The title for 2.3.2 should be “Scanning electron microscopy (SEM) and energy dispersive X-ray spectroscopy (EDS)”

subtitle changed as noted by reviewer.

In 3, all subtitle should be 3. Not 4..

subtitle number changed as noted by reviewer.

In 3, -CH2 should be -CH2.

the change was made as directed by the reviewer

In Figure 3, why some blue lines are not at the peak position?

In the transmittance mode FTIR-ATR spectrum, the peaks that mark the minima are studied. In RAMAN analysis, the peaks are studied at their maximums. In this work, the FTIR analysis was applied to study the functional groups because the molecules are polar and we had the equipment in the laboratory.  Some blue lines are corrected so that they indicate the minimums as indicated by the reviewer.

In 3.1, “The stretching bonds of the Si-O-C and Si-O-Si groups were assigned at 1083 and 1047 cm-1”. These numbers are different from those shown in Figure 4.

The change of signal values ​​was made as indicated by the reviewer(1085 and 1000 cm-1).

In Table 3, what do the orange color mean?

I removed the orange color in the spaces. What I indicated there is that the area under the curve could not be quantified, therefore it is empty.

In 3.1, “…at 1600 cm-1 is the double bond C=C stretching, at 12 cm-1 are the stretching vibrations of C-C bond…” Where is the number 12 from?

the correction of 12 cm-1 by 1200 cm-1 was made

In 3.2, “Figure 5 shows the SEM images of the three different graphene oxides…” should be “Figure 6 shows the SEM images of the three different graphene oxides…”.

The correction indicated by the reviewer was made.

The format for figure number is not consistent, like Figure 6[a), Figure 8 (a and b), Figure 9 a, b…

The correction indicated by the reviewer was made.

In 3.2.2, “This phenomenon, would promote the transference o…” What is o?

The correction indicated by the reviewer was made.

In Figure 11, “Resina epoxy” should be “Resin epoxy”.

The correction indicated by the reviewer was made.

In 3.4, please add details about force-displacement curves and how authors calculated Young’s modulus and resistance to traction.

In figure 13 a), the samples with an increase in their mechanical properties under tension are shown.

Young's modulus was calculated through load increment and the extension, between two points on the line as far apart as possible, using the following equation:

Young's modulus is calculated by means of load increment and the extension, between two points on the line as far apart as possible, Where Y is applied axial stress and X strain data, using the following equation:

E=((∑▒〖XY-KX Ì…Y Ì… 〗)/(∑▒〖X^2-KX Ì…^2 〗))

In 3.4, “…exhibited values higher than…” should be “…exhibited values higher…”.

The correction indicated by the reviewer was made.

In 3.4, please provide evidences for the statement “However, a load above 1% weight GO evidences differences between Young’s modulus and a minor resistance to traction between GO/epoxy (3% weight GO) and GO-GPTMS/epoxy (3% weight GO).

In 3.4, “…with the increase in peak deformation regarding traction by 11 and 51%...” How did authors calculate this? The degree of oxidation was not mentioned.

Round 2

Reviewer 1 Report

Dear Editor and Authors,

firstly I want to underline that the manuscript has been improved, but some need also some modifications:

1. In Fig 2, please use the English language.

2. Please add a method to characterize the modified GO, for example, XPS, if Raman is not appropriate.

Author Response

The method we apply (FTIR) is appropriate for characterization. We do not use Raman because the characterized molecules have a dipole moment. XPS is a technique to measure binding energies and it would be interesting in future work to carry out work that aims to study binding energies, but at the moment we do not have the equipment and we would have to establish a new project to do so.

Reviewer 3 Report

The manuscript was revised according to the reviewer's comments and I recommend it to be published in Journal of Carbon Research.

Author Response

The requested corrections were made, both in the mechanical properties, adding the stress-strain curves, as well as the corrections regarding FTIR and the corresponding clarifications.

Round 3

Reviewer 1 Report

accept